# Deep Correlated Prompting for Visual Recognition with Missing Modalities

**Lianyu Hu,Tongkai Shi, Wei Feng, Fanhua Shang, Liang Wan**[*]
College of Intelligence and Computing, Tianjin University
{hly2021,stk,wfeng,fhshang,lwan}@tju.edu.cn
https://github.com/hulianyuyy/Deep_Correlated_Prompting

## Abstract

Large-scale multimodal models have shown excellent performance over a series of tasks powered by the large corpus of paired multimodal training data. Generally, they are always assumed to receive modality-complete inputs. However, this simple assumption may not always hold in the real world due to privacy constraints or collection difficulty, where models pretrained on modality-complete data easily demonstrate degraded performance on missing-modality cases. To handle this issue, we refer to prompt learning to adapt large pretrained multimodal models to handle missing-modality scenarios by regarding different missing cases as different types of input. Instead of only prepending independent prompts to the intermediate layers, we present to leverage the correlations between prompts and input features and excavate the relationships between different layers of prompts to carefully design the instructions. We also incorporate the complementary semantics of different modalities to guide the prompting design for each modality. Extensive experiments on three commonly-used datasets consistently demonstrate the superiority of our method compared to the previous approaches upon different missing scenarios. Plentiful ablations are further given to show the generalizability and reliability of our method upon different modality-missing ratios and types.

## 1  Introduction

Our human beings typically perceive information of multiple modalities such as visual, linguistic and audio signals to understand the world, where different signals drawn from various perspectives are inherently complementary. Thus, modeling and coordinating multimodal information is of great value for large-scale models to reason about real-world scenarios. Recently, multimodal models [34, 1, 22, 29] have developed fast powered by the large collection of multimodal data pairs and evolution of model architectures (e.g., Transformer [31]), which demonstrate promising performance across a series of downstream tasks, such as cross-model retrieval [8, 18, 29, 9], image captioning [24, 1] and image/video generation [23, 22]. Supported by their large capacity and general knowledge acquired by training upon web-scale data, these models have been more and more applied to daily work in our life (e.g., GPT-4 is used by millions of people).

However, there exist two major concerns that may hinder these impressive models from broader applications. First, a common assumption of previous methods is that the input data is modality-complete, which may not always hold in the real world due to privacy considerations, collection difficulty and security issues [27, 19]. When an input modality is missing in general real-world conditions, the performance of these models usually degrades a lot (regardless of training or testing settings) [27], which is easily influenced by the input completeness. Second, these large models are usually parameter-abundant [1, 6, 21] and require heavy computations [30, 45] to pretrain and

---

[*]Corresponding author

38th Conference on Neural Information Processing Systems (NeurIPS 2024).

finetune on downstream tasks, whose computational demands may not always be available in most real-world applications due to limited computing resources. It's necessary to develop a new method to efficiently adapt these powerful methods to perform robustly against missing-modality scenarios.

Previous works [27, 42, 32, 28, 19] of multimodal learning have considerably explored the missing-modality issues. Earlier works [27, 42, 32, 28] mostly reconstruct the absent information of missing modalities or use other modalities to augment the missing modalities. MMP [42] has first introduced prompt learning to handle missing-modality scenarios by regarding different missing cases as different types of input. The multimodal backbone is kept frozen and only the newly introduced prompts are updated in the fine-tuning process, thus only incurring a few extra computations. However, MMP [19] simply inserts independent prompts into each layer, and overlooks the relationships among prompts and input features. The prompts across different layers lack cooperation to aggregate beneficial information to well guide the model predictions. The fixed prompts are used for different input samples which fail to consider the characteristics of various inputs. The complementary multimodal information of different input modalities is overlooked in the fine-tuning process.

To better adapt large multimodal models to missing-modality scenarios, we introduce deep correlated prompting (DCP) by capturing different types of correlations between prompts and input features. Specifically, to leverage the hierarchical semantics of different layers, we propose correlated prompts by perceiving beneficial information from preceding layers to instruct the features of the current layer. We further propose to dynamically generate the prompts according to the input features to better fit the characteristics of different inputs. To leverage the complementary information of multimodal inputs, we decompose the prompts into modal-common and modal-specific parts to guide each encoder to focus on its unique features. The proposed prompts are concatenated and prepended to the input and intermediate features of the multimodal backbone to instruct the model to alleviate the performance drop caused by the missing modality. The multimodal backbone keeps frozen during training and only the learnable prompts are tuned, thus offering high training efficiency. Extensive experiments on three widely-used datasets including MM-IMDb [2], UPMC Food-101 [33] and Hateful Memes [17] demonstrate consistently superior performance compared to other methods across all benchmarks, which verify the effectiveness of our proposed method. Ablation studies are further given to verify the generalizability and reliability of our method upon different missing-modality types and ratios.

## 2    Related Work

### 2.1    Missing-Modality for Multimodal Learning.

Earlier works of missing modalities for multimodal learning mostly generate the missing modality based on other modalities [42], or align features of latent representations for multiple modalities to help recognition [38, 15]. Recently, multimodal transformers [1, 22, 23] emerge as an effective tool to model information from various modalities and process them into a robust representation, which have shown impressive performance over a series of tasks [8, 18, 29, 9, 24, 11, 12]. However, these methods typically assume that the inputs are modality-complete, which may not always hold in real-world scenarios. When a modality is missing, these methods usually demonstrate degraded accuracy and lead to unstable performance [27].

To deal with missing modalities in multimodal transformers, MMIN [42] predicts the intermediate features of the missing modality based on other available modalities, by learning a common multimodal representation. SMIL [28] proposes a Bayesian meta-learning framework to estimate the latent features of the modality-incomplete data. It further explores the effects faced with severe modality-incomplete samples (e.g., 90% missing ratio). Ma et al. [27] test the robustness of multimodal transformers to missing modalities and their missing types, and propose a multitask optimization framework to improve it via modality fusion. Zeng et al. [40] propose a tag-assisted transformer Encoder network to handle the problem of missing uncertain modalities. ShaSpec [32] employs a shared head across different tasks to aggregate information from various input samples, which complements the features for missing modalities. However, these methods mostly assume that a fixed modality is missing, which is known in advance. Besides, these methods still require updating most parameters of the model, which consume heavy computations in downstream tasks. MMP [19] first introduced missing-aware prompts to handle scenarios with missing modalities with minimal extra computational costs. However, it simply inserts independent prompts into the intermediate features of the multimodal backbone, without considering the relationships between prompts of different

layers as well as the correlations between prompts and input features. In contrast, we carefully design the prompts by exploring the correlations between prompts and input features, and achieves much superior performance across all missing-modality scenarios upon three commonly-used datasets.

## 2.2 Prompt Learning

Prompt Learning is first explored in neural language processing (NLP), which adopts a "prompt" to modify the input text to instruct the pre-trained model for downstream tasks. Earlier works [6, 14] usually adopt manually designed prompts to improve the generalizability of large models over downstream tasks. Later, prompt tuning methods [26, 20, 25] emerge by prepending learnable prompts to the input features in the training phase to automate the optimization process. Recently, prompt learning is also introduced into computer vision tasks [4, 13, 37] and multimodal learning tasks [36, 44, 35, 16, 43]. CoOp [44] is first proposed to insert learnable soft prompts besides input images to adapt vision-language models to various vision tasks. CoCoOp [43] generates an image-conditional prompt to utilize the power of input features. ProGrad [46] only updates the prompts whose gradients are aligned to the "general knowledge" generated by the original prompts. KgCoOp [36] tries to align the output embeddings of the text encoder with those of the pretrained CLIP to preserve beneficial information. MaPLe [16] injects deep learnable soft prompts into both the image and text branches to enable prompt collaboration. DualPrompt [35] extends the prompts to learn different task information conditionally in continual learning. DePT [41] decouples base-specific knowledge from feature channels into an isolated feature space during prompt tuning. These works have demonstrated the effectiveness of prompt learning to adapt large-scale vision-language models for downstream tasks with minimal extra computation costs. In this paper, we introduce prompt learning into multimodal models to increase their robustness to missing-modality scenarios, via attaching different types of prompts according to various missing cases.

## 3 Method

### 3.1 Overall framework

**Problem definition.** We first provide a succinct overview of the missing-modality scenario addressed in this paper. Without loss of generalizability, we explore multimodal inputs with $M = 2$ modalities $m_1$ and $m_2$ (e.g., text and image), which is naturally extensible to more modalities. Specifically, with a multimodal dataset $D = \{D^c, D^{m_1}, D^{m_2}\}$, we denote $D^c = \{x^{m_1}, x^{m_2}, y\}$ as the modality-complete case, where $x$ denotes the input and $y$ represents the label. We let $D^{m_1} = \{x^{m_1}, y\}$ and $D^{m_2} = \{x^{m_2}, y\}$ denote the modality-incomplete cases, wherein one modality is absent (e.g., missing text and missing image). Some examples of modality-complete and modality-incomplete inputs are depicted in the lower part of Fig. 1.

**Recap MMP [19].** MMP [19] first introduced missing-aware prompts to handle missing-modality cases. Specifically, it assigns $2^M - 1$ types of prompts for tasks involving $M$ modalities (e.g., 3 prompts for vision-language tasks, including one prompt for modality-complete case, one prompt for image-only case, and one prompt for text-only case). Given an input sample, it first selects the prompts corresponding to the missing case and then prepends the prompts to the input and intermediate features of the multimodal backbone, instructing the pretrained model to perform prediction. In this procedure, only the learnable prompts are updated and the multimodal backbone keeps frozen.

### 3.2 Overall framework

Though MMP [19] has achieved notable progress in improving the robustness of multimodal models upon missing-modality cases by incurring minimal computational costs, it only assigns independent prompts to the input and intermediate features, which (1) fails to consider the relationships between prompts of different layers, and (2) lacks the correlations between prompts and input features, and (3) overlooks the complementarity of multimodal inputs. To better adapt the pretrained multimodal model for missing-modality scenarios, we propose to design three types of missing-aware prompts by capturing the relationships between prompts and inputs. First, we generate missing-aware prompts by leveraging the correlations of preceding prompts across multiple layers and various modalities. Second, we dynamically generate the prompts for each input sample to fit its characteristics. Third,

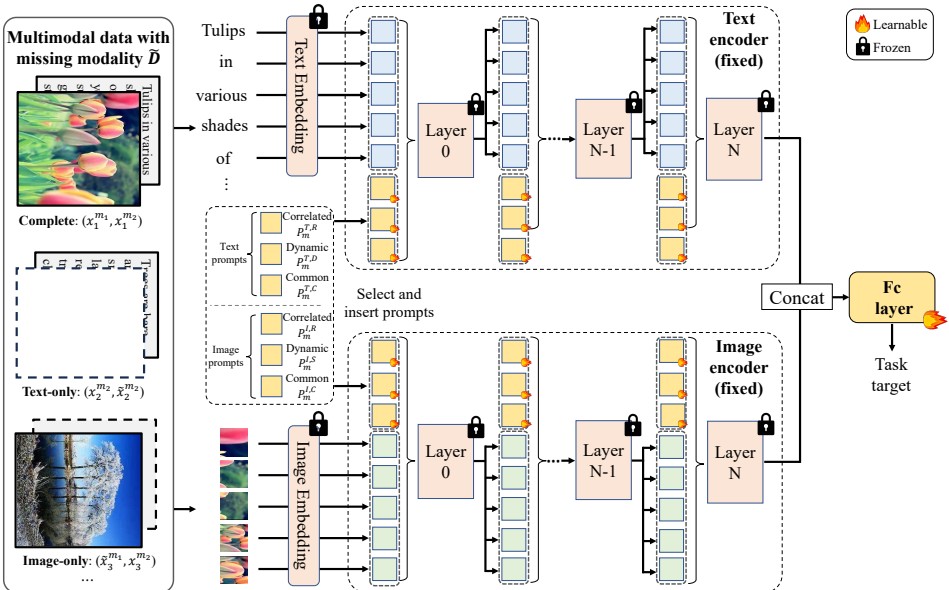

Figure 1: The overview of our proposed framework. We first select the prompt $P_m^T$ and $P_m^I$ with $m \in \{c, m_1, m_2\}$ for the text encoder and image encoder according to the missing case (e.g., complete, text-only, image-only) of the multimodal inputs ($x^{m1}$, $x^{m2}$). The prompt $P_m^T$ ($P_m^I$) is composed of three types of missing-aware prompts including the correlated prompts $P_m^{T,R}$ ($P_m^{I,R}$), dynamic prompts $P_m^{T,D}$ ($P_m^{I,D}$) and modal-common prompts $P_m^{T,C}$ ($P_m^{I,C}$). Then we prepend the prompts to the inputs and intermediate features of both encoders to instruct the model to fit the missing case. Finally, we concatenate the task-related token of both encoders as the final representation, and pass it through a fully-connected layer for class prediction. In the whole procedure, only the fully-connected (fc) layer and deep correlated prompts are updated while others keep frozen.

we decompose multimodal prompts into modal-common parts and modal-specific parts, which fuses beneficial information from other modalities and enables each encoder to focus its unique features.

Fig. 1 shows the framework overview of our proposed method. Specifically, without loss of generalizability, we adopt the widespread two-stream multimodal method CLIP [29] as our backbone. Given the input text and image, we first employ pretrained text and image embedding layers from the pretrained CLIP [29] to convert them into token sequences. For each input sample, we select the corresponding prompt $P_m^T$ and $P_m^I$ for the text encoder and image encoder, respectively, given the type of missing modality with $m \in \{c, m_1, m_2\}$. The prompt for each encoder is composed of three types of missing-aware prompts including the correlated prompts $P_m^{T,R}$ ($P_m^{I,R}$), dynamic prompts $P_m^{T,D}$ ($P_m^{I,D}$) and modal-common prompts $P_m^{T,C}$ ($P_m^{I,C}$). We concatenate the missing-aware prompts and prepend them to the input tokens $x^{m_1}$ ($x^{m_2}$) as a whole sequence to process. Finally, we concatenate the task-related token of both encoders as the final output representation, and pass it through a fully-connected layer for task prediction. In this procedure, only the parameters of the fully-connected layer and the newly introduced deep correlated prompts are updated in the training procedure, and the backbone (i.e., the text embedding layer, image embedding layer, text encoder and image encoder) is kept frozen. The illustration of our proposed prompts is given in Fig. 2. We next introduce our prompting designs in detail.

### 3.3 Deep Correlated Prompt Learning

#### 3.3.1 Correlated Prompts

MMP [19] append independent prompts to the input and intermediate features of the multimodal backbone to guide model predictions. Though it could theoretically provide enough guidance for features in each layer, the prompts across each layer lack synergy, which fails to cooperate with the representations of each layer that various semantics. We argue that prompts across consecutive layers should be closely correlated to receive necessary semantics from preceding layers to instruct

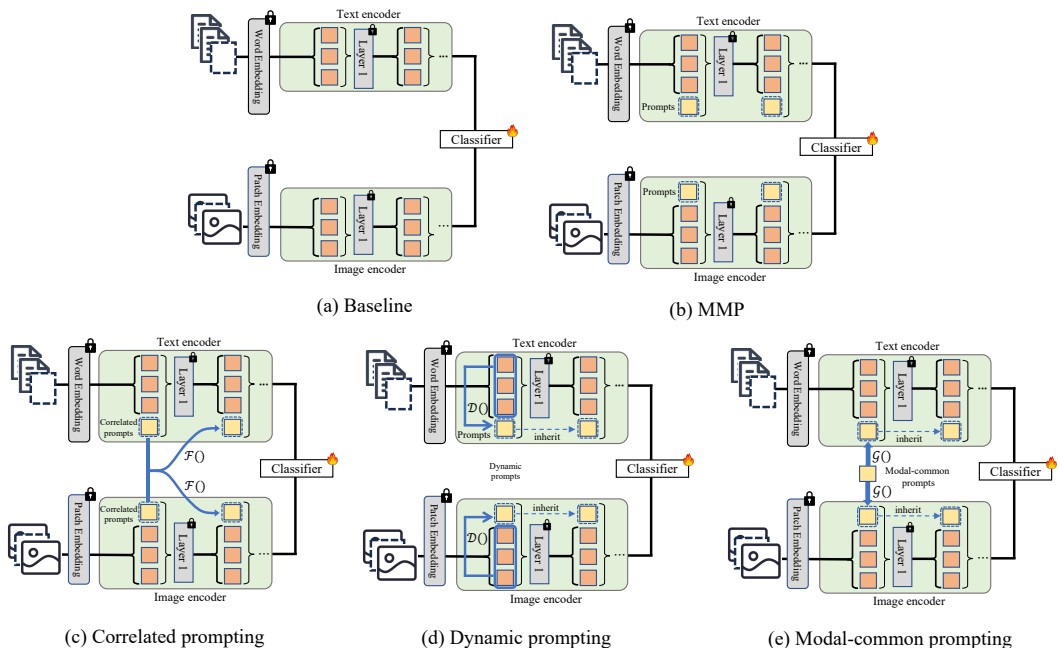

Figure 2: (1) Baseline, which simply uses fixed image encoder and text encoder and only finetunes the classifier to handle downstream tasks. (2) MMP, which inserts independent prompts at each layer to guide the model to handle missing-modality cases. (3) Correlated prompts, which generate the prompts of the next layer based on the prompts of both modalities in the current layer to enable cooperation of prompts from both modalities. (4) Dynamic prompts, which dynamically computes the prompts based on different input features to better guide the behavior of the model, avoiding using fixed prompts for different inputs. (5) Modal-common prompts, which store the shared information across different modalities and facilitate the model to encode modal-specific information to better handle the missing scenarios in each modality.

the current layer to handle missing modalities. Thus, we propose to generate the prompts of the current layer based on the observation of the preceding prompts. Besides, the features of different modalities usually contain complementary information. Leveraging the complementary information from various modalities could further help the model predictions. We propose to incorporate the beneficial information from multimodal inputs to help guide the outputs of each modality.

Specifically, we prepend the prompts to the input features and the intermediate features of the multimodal backbone up to a depth of $J$. Taking the image encoder as an example, the calculation process of the $i_{th}$ ($i \in [0, \ldots, J-1]$) layer $\mathcal{V}_i$ could be expressed as:

$$[\_\_, X_i^I] = \mathcal{V}_i([P_{m,i-1}^{I,R}, X_{i-1}^I]). \tag{1}$$

Here, $X_i^I$ is the features of the $i_{th}$ layer in the image encoder and $P_{m,i-1}^{I,R}$ is the newly introduced correlated prompts of the $(i-1)_{th}$ layer. After the $J_{th}$ layer, the correlated prompt is retained from the previous layer, and the calculation process for the $i_{th}$ ($i \in [J, \ldots, N-1]$) layer $\mathcal{V}_i$ can be presented as:

$$[P_{m,i}^{I,R}, X_i] = \mathcal{V}_i([P_{m,i-1}^{I,R}, X_{i-1}^I]) \tag{2}$$

where $N$ denotes the length of all layers.

To leverage the correlations of prompts between different layers, we generate the prompts of the $i_{th}$ ($i \in [1, \ldots, J-1]$) layer $\mathcal{V}_i$ by observing prompts of its preceding layer $\mathcal{V}_{i-1}$ to inject beneficial information, as:

$$P_{m,i}^{I,R} = \mathcal{F}_{i-1}^I(P_{m,i-1}^{I,R}) \tag{3}$$

where $\mathcal{F}_{i-1}^{I,R}(\cdot)$ denotes the prompt generation function for the $(i-1)_{th}$ layer in the image encoder. To decrease the required parameters and add non-linearity to $\mathcal{F}(\cdot)$, we adopt $\mathcal{F}(\cdot)$ as a bottleneck

MLP with the GELU activation [10] in between, followed by a LayerNorm (LN) function [3] as:

$$\mathcal{F}(\cdot) = \text{LN}(\text{Fc}(\text{GELU}(\text{Fc}(\cdot)))). \tag{4}$$

The intermediate feature dimension of $\mathcal{F}(\cdot)$ is $r$ (usually $r = \frac{1}{16}$) times of its input channel dimension. For the correlated prompts at the input level, we leave it randomly initialized without the generation procedure.

To leverage complementary information from different modalities, we incorporate prompts from both encoders to fuse their distinct semantics for instructions. Taking the image encoder as an example, we generate the prompts of the $i_{th}$ ($i \in [1, \ldots, J-1]$) layer $\mathcal{V}_i$ based on the prompts of the layer $\mathcal{V}_{i-1}$ in the image encoder and the prompts of layer $\mathcal{T}_{i-1}$ in the text encoder as:

$$P_{m,i}^I = \mathcal{F}_{i-1}^I(\text{Concat}(P_{m,i-1}^I, P_{m,i-1}^T)). \tag{5}$$

Here, Concat denotes the concatenation operation. Instead of leaving prompts uncorrelated across different modalities, our design improves mutual synergy between different modalities to help adapt to various missing scenarios.

### 3.3.2 Dynamic prompts

Different input samples usually contain information of various semantics. Using fixed prompts for different inputs may not well fit their distinct features, and thus can't offer enough guidance for the model to fit different missing-modality scenarios. Thus, we propose to dynamically generate the prompts based on the input features to adjust the instructions to fit the characteristics of different inputs.

Specifically, taking the image encoder as an example, we generate the dynamic prompts $P_m^{I,D}$ based on the input features $X_0^I$ as:

$$P_m^{I,D} = \mathcal{D}^I(X_0^I) \tag{6}$$

where $\mathcal{D}^I$ denotes the dynamic prompt generation function for the image encoder. To deal with the varying length of tokens for different inputs, we instantiate $\mathcal{D}^I$ as a self-attention layer [31], whose architecture can be expressed as :

$$\mathcal{D}(\cdot) = \text{LN}(\text{MLP}(\text{LN}(\text{MHA}(\cdot)))). \tag{7}$$

Here, MHA is the multi-head attention mechanism in transformers [31]. We set the number of heads as 1 for simplicity. The dynamic prompts are only inserted at the input level. For the subsequent layers, the prompts are retained from the previous layers and updated together with intermediate features following Eq. 2.

### 3.3.3 Modal-common prompts

Multimodal inputs usually contain both modal-specific information and modal-common information across different modalities. By disentangling the modal-common features from the modal-specific features, each modality could aggregate beneficial information from the shared features across various input modalities and build more powerful representations based on its unique characteristics. We decompose the multimodal missing-aware prompts into modal-common parts and modal-specific parts to store common characteristics across different modalities and model-specific features for each modality, respectively.

Specifically, we introduce a modal-common prompt $P_m^C$ for multimodal inputs to embody the mutual features across different modalities, and accordingly encourage the proposed correlated prompts and dynamic prompts to embed modal-specific instructions. To cooperate with the features of different modalities, we project the modal-common prompt $P_m^C$ to the image and text space to obtain $P_m^{T,C}$ and $P_m^{I,C}$ via projection layers, respectively, as:

$$\begin{aligned} P_m^{T,C} &= \mathcal{G}^T(P_m^C) \\ P_m^{I,C} &= \mathcal{G}^I(P_m^C). \end{aligned} \tag{8}$$

Here, $\mathcal{G}^T$ and $\mathcal{G}^I$ are the projection layers for the text modality and the image modality, respectively, which are both instantiated as a MLP with the intermediate feature reduction factor $r$=16. We only insert modal-common prompts at the input level and leave the prompt in the intermediate layers reserved and updated following Eq. 2.

# 4 Experiments

## 4.1 Experimental Setup

**Datasets.** We follow the previous works [19, 27] to evaluate our methods:

MM-IMDb [2] is currently the largest publicly available multimodal dataset for genre prediction on movie genre classification. It is notated with both image and text modalities for 25959 movies. As a movie might have several genres, the genre prediction task is thus a multi-label classification task.

UPMC Food-101 [33] is a large multimedia dataset consisting of noisy image-text pairs collected from Google Image Search with 101 food categories. It has identical categories to the largest publicly available ETHZ Food-101 dataset [5], used to classify the categories of different foods.

Hateful Memes [17] is a multimodal dataset for hateful meme detection (image + text) that contains 10,000+ new multimodal examples created by Facebook AI. To prevent the model from relying on a single modality, it is constructed to make unimodal models more likely to fail.

**Implementation details.** We use CLIP [29] as our multimodal backbone, with ViT-B/16 [7] as the image encoder. For the image input, we follow CLIP [29] to resize input images into $224 \times 224$. The patch size is set as 16 for the multimodal transformer. For the text modality, we use the tokenizer from pretrained CLIP to tokenize the text input. The maximum length of text inputs is 77. We freeze all the parameters of both the image encoder and text encoder, and only tune the parameters of deep correlated prompts and the fc layer (for task target). We set the length $L_p$ of learnable prompts as 36 and prepend them to the features of $M = 6$ layers. We use Adam optimizer with initial learning rate of 1e-2 and weight decay 2e-2. The learning rate is warmed up for 10% of the total training steps and then is decayed linearly to zero. We perform our experiments with batch size of 4 on a 3090 GPU. For the missing modality, we stop feeding the inputs into the corresponding encoder and use a zero-filled tensor as the output instead.

**Metrics.** We use corresponding proper metrics for each dataset to evaluate our method. For MM-IMDb [2], we adopt F1-Macro to measure the multi-label classification performance. For UPMC Food-101 [33], we employ the top-1 classification accuracy to evaluate the recognition performance. For Hateful Memes [17], we use Area Underthe Receiver Operating Characteristic Curve (AUROC).

**Setting of Missing Modality.** In this paper, we focus on the general realistic scenarios in real life, where any modality may appear in both training and testing phases. To stimulate this condition, we follow MMP [19] to define the missing rate $\eta$ as the proportion of modality-incomplete data to the entire dataset. For vision-language tasks, there exist three types of missing cases: missing-both (text and image), missing-text and missing-image. For the missing-both case, the training and testing data are composed of $\frac{\eta}{2}$ text-only data, $\frac{\eta}{2}$ image-only data and (1-$\eta$) modality-complete data. For the missing-text and missing-image cases, the training and testing data are composed of $\eta$ image-only (text-only) data and (1-$\eta$) modality-complete data. This definition could be naturally extended to data with more modalities by using ( $\frac{\eta}{M^2-2}$) modality-incomplete data for each missing case and (1-$\eta$) complete data. In our experiments, we use $\eta$=70% by default.

## 4.2 Experimental Results

**Effectiveness.** We first verify the effectiveness of our proposed components across different missing cases including missing-image, missing-text and missing-both. We conduct the experiments upon the MM-IMDb [2] dataset with missing rates ranging from 0% to 100%. The methods used for comparison include (1) baseline, which directly sets the features as zeros when a modality is missing; (2) Ours (A), which only equips the correlated prompts; (3) Ours (B), which equips both the correlated prompts and the dynamic prompts; (4) Ours, which uses all the three proposed prompts. The only difference between our method and the baseline is inserting learnable prompts at each layer which only bring few extra parameters. The results are shown in Fig. 3.

It's first noticed that compared to the baseline, all our three variants could notably promote the performance across all missing rates for various missing-modality cases, demonstrating strong robustness to different missing-modality scenarios. Using the correlated prompts boosts the performance most, and equipping the dynamic prompts or modal-common prompts offers a similar performance boost. Using all three proposed prompts achieves the best performance, which verifies the effectiveness of

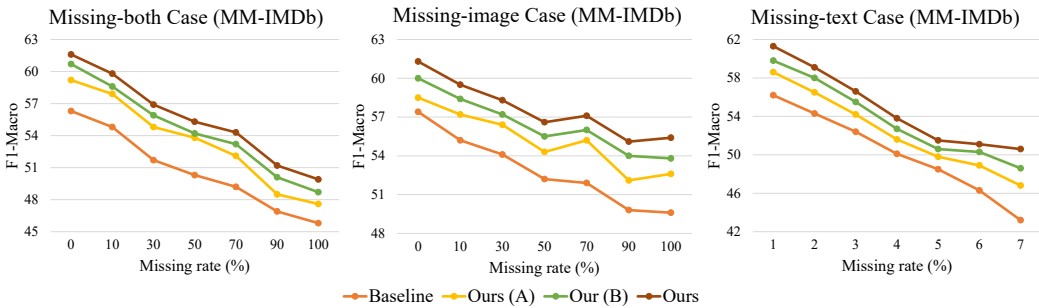

Figure 3: Comparison of our final model (Ours) with (1) baseline, which directly drops the features when a modality is missing; (2) Ours (A), which only equips the correlated prompts; (3) Ours (B), which equips both the correlated prompts and the dynamic prompts. The experiments are conducted on the val set of MM-IMDb dataset [2] across different missing rates (0–100%) upon three different missing-modality scenarios (missing-both, missing-image and missing-text).

capturing correlations between prompts and input features. An interesting observation is that the performance degradation is much smaller when input images are missing compared to missing-both and missing-text, which shows that text is more important for the task target on this dataset.

**Ablation study.** We test the configurations for our proposed three prompts on the val set of MM-IMDb dataset in Tab. 1, Tab. 2 and Tab. 3, respectively. In the upper part of Tab. 1, we first explore how to leverage the correlations between prompts of different layers. "No projection" denotes using independent prompts for each layer and "Fc" denotes using a fully connected layer for projection. It's observed that compared to using independent prompts, either using a fc or a MLP offers a notable performance boost. The MLP gives better performance for its non-linearity and we use it by default. We then explore the prompt depth $J$ (how many layers to insert prompts) in the middle of Tab. 1. We notice that the performance reaches a peak when $J$=6, and either decreasing it or increasing it would degrade the performance. Finally, we verify the efficacy of incorporating multimodal prompts for instruction in the bottom part of Tab. 1. It's observed that introducing bi-modal prompts to generate the prompts of the current layer gives better performance, which shows that different modalities can offer complementary information.

We test the configurations for the dynamic prompts in Tab. 2. We first explore the prompt depth for the dynamic prompts. It's observed that the performance reaches a peak when the prompt depth equals 1, and continues to decrease when the prompt depth increases. We thus set the prompt depth as 1. We then explore the generation functions for the dynamic prompts. Besides the default attention mechanism, we test other alternatives which use a maximum, minimum or average pooling layer followed by a fc layer to generate the prompts. It's observed that all approaches could give a notable performance boost, and the attention function achieves the best performance.

Table 1: Ablations for the correlated prompts.

| Configurations | F1-Macro(%) |
|---|---|
| No projection | 52.54 |
| Fc | 53.76 |
| MLP (r=16) | **54.24** |
| Depth = 3 | 53.74 |
| Depth = 6 | **54.24** |
| Depth = 12 | 53.86 |
| uni-modal | 53.52 |
| bi-modal | **54.24** |

Table 2: Ablations for the configurations of dynamic prompts.

| Configurations | F1-Macro(%) |
|---|---|
| Depth = 1 | **54.24** |
| Depth = 2 | 54.08 |
| Depth = 3 | 53.76 |
| Depth = 6 | 53.52 |
| Attention | **54.24** |
| Max & Projection | 53.12 |
| Min & Projection | 53.35 |
| Avg & Projection | 53.42 |

Table 3: Ablations for the modal-common prompts.

| Configurations | F1-Macro(%) |
|---|---|
| Depth = 1 | 54.24 |
| Depth = 2 | 54.16 |
| Depth = 3 | 54.42 |
| Depth = 6 | **54.52** |
| Fc | 53.76 |
| MLP (r=4) | **54.36** |
| MLP (r=8) | 54.12 |
| MLP (r=16) | 54.24 |

We explore the configurations for the modal-common prompts in Tab. 3. We first examine the prompt depth for the modal-common prompts. It's observed that as the prompt depth ranges from 1 to 6, our model achieves similar results and reaches a peak when the prompt depth equals 6. Considering the accuracy-computation trade-off, we set the prompt depth as 1 by default. We then explore the projection functions for the modal-common prompts. Generally, we observe that using a MLP for

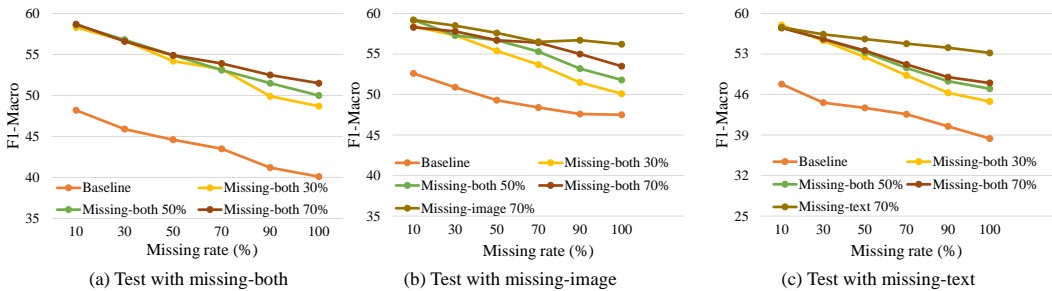

Figure 4: Ablations on the generalizability to different testing scenarios across various missing rates on the val set of MM-IMDb dataset [2]. (a) All models are trained on *missing-both* cases, and evaluated on *missing-both* cases with different missing rates. (b) Models are trained on *missing-both* or *missing-image* cases, and evaluated on *missing-image* cases with different missing rates. (c) Models are trained on *missing-both* or *missing-text* cases, and evaluated on *missing-text* cases with different missing rates.

projection performs better than using a fc for projection. Using different channel reduction factors $r$ including 4, 8 and 16 achieves similar performance. We thus use a MLP with $r$=16 by default.

**Generalizability.** We conduct experiments to verify the generalizability of our method when trained and tested upon different missing cases. We evaluate models trained on *missing-both* cases or *missing a specific modality*, and test them on the missing-both, missing-image and missing text cases in Fig. 4(a), (b) and (c), respectively. It's first noticed that all our variants outperform the baseline by a large margin across various missing cases and missing rates. Models trained with a certain modality missing (e.g., missing-text, missing-image), usually perform slightly better than models trained upon missing-both cases when tested with a certain modality missing. Models trained with missing both modalities perform robustly to cases with any missing type (e.g. missing-both, missing-text and missing-image). We further observe the model trained with higher missing rates (e.g., 70%), is more robust to high-rate missing cases (70%–100%) during testing than models trained with lower missing rates (e.g., 30% and 50%). It is concluded that models trained on missing-both cases are robust to various missing cases, which demonstrate strong robustness compared to training with one modality.

**Comparison with other methods.** We compare our method with recent approaches upon three commonly-used datasets, i.e., MM-IMDb [2], UPMC Food-101 [33] and Hateful Memes [17], to verify its effectiveness upon different missing-modality scenarios. We include the following methods for comparison (1) baseline, which directly drops the features when a modality is missing; (2) CoOp [44], which only prepends prompts at the input level; (3) MMP [19], which inserts independent prompts for the input and intermediate features of the multimodal backbone; (4) MaPLe [16], which generates the prompts in the image encoder based on those of the text encoder; (5) DePT [41], which decouples base-specific knowledge from feature channels into an isolated feature space during prompt tuning. MMP [19] is originally based on the ViLT backbone and we reimplement it following the same protocol as ours. We compare these methods across three different missing rates including $\eta$=50%, $\eta$=70% and $\eta$=90% upon three various missing-modality cases including missing-image, missing-text and missing-both in Table. 4. Our method largely outperforms other methods on three datasets across different missing rates, which verifies its robustness to different missing scenarios. Compared to MMP [19], our method exceeds it over all missing cases by a large margin, which demonstrates the superiority of leveraging the correlations between prompts and input features.

An interesting observation is that on most datasets (e.g., MM-IMDb [2] and Food101 [33]), missing text input usually has higher effects on the performance than missing image input. We figure that texts are more crucial for the tasks by providing more detailed explanations and precise captions than images on these two datasets. Instead, on the Hateful Memes dataset [17], it's observed that missing image input influences the results more than missing text or missing both. This may reflect the different focus when constructing various datasets.

**Efficiency.** Compared with only finetuning the last fc layer to fit downstream tasks, our deep correlated prompts only introduce extra 4.0M parameters, which just consume 2.4% of the entire model (151M), but could notably improve the performance by a large margin over different missing cases and missing rates.

Table 4: Comparison with CoOp [44], MMP [19], MaPLe [16] and DePT [41] on the MM-IMDb [2], UPMC Food-101 [33], and Hateful Memes [17] datasets under various missing-modality cases with different missing rates. The bold number indicates the best performance.

| Datasets | Missing rate $\eta$ | Train/Test Image | Text | Validation set CoOp | MMP | MaPLe | DePT | Ours | Testing set CoOp | MMP | MaPLe | DePT | Ours |
|---|---|---|---|---|---|---|---|---|---|---|---|---|---|
| MM-IMDb (F1-Macro) | 50% | 100% | 50% | 51.23 | 52.07 | 52.76 | 53.87 | **55.23** | 48.06 | 48.88 | 49.58 | 50.64 | **52.13** |
| | | 50% | 100% | 53.04 | 54.52 | 55.26 | 56.04 | **57.32** | 49.89 | 51.46 | 52.32 | 52.78 | **54.32** |
| | | 75% | 75% | 51.46 | 52.12 | 52.87 | 54.02 | **55.45** | 48.37 | 49.32 | 49.56 | 50.87 | **52.32** |
| | 70% | 100% | 30% | 47.26 | 48.23 | 48.75 | 49.87 | **51.35** | 44.13 | 45.64 | 45.52 | 46.38 | **48.52** |
| | | 30% | 100% | 52.32 | 53.21 | 53.98 | 55.04 | **56.21** | 48.82 | 50.52 | 50.64 | 52.13 | **53.14** |
| | | 65% | 65% | 50.22 | 51.34 | 52.31 | 53.17 | **54.24** | 46.84 | 48.12 | 49.16 | 50.32 | **51.42** |
| | 90% | 100% | 10% | 47.86 | 48.84 | 50.12 | 50.98 | **52.36** | 44.76 | 45.32 | 46.84 | 47.56 | **49.26** |
| | | 10% | 100% | 51.65 | 52.36 | 53.14 | 54.12 | **55.42** | 48.32 | 49.12 | 50.13 | 50.88 | **52.22** |
| | | 55% | 55% | 47.44 | 48.04 | 48.82 | 49.98 | **51.26** | 44.12 | 44.87 | 45.12 | 46.54 | **48.04** |
| Food101 (Accuracy) | 50% | 100% | 50% | 77.36 | 78.24 | 79.87 | 80.24 | **82.33** | 77.45 | 77.89 | 79.64 | 80.16 | **82.11** |
| | | 50% | 100% | 86.98 | 87.12 | 87.48 | 87.85 | **89.23** | 87.02 | 87.16 | 87.35 | 82.14 | **89.12** |
| | | 75% | 75% | 81.76 | 81.98 | 82.58 | 83.26 | **85.25** | 81.24 | 81.72 | 82.34 | 83.12 | **85.24** |
| | 70% | 100% | 30% | 76.65 | 76.74 | 76.87 | 76.87 | **79.18** | 76.34 | 76.52 | 77.02 | 77.34 | **78.87** |
| | | 30% | 100% | 85.21 | 86.12 | 86.36 | 86.52 | **87.53** | 84.78 | 85.64 | 85.89 | 86.12 | **87.32** |
| | | 65% | 65% | 79.14 | 79.56 | 80.06 | 81.85 | **82.38** | 78.87 | 79.12 | 79.84 | 81.46 | **81.87** |
| | 90% | 100% | 10% | 72.65 | 73.74 | 73.25 | 74.22 | **75.54** | 71.87 | 73.14 | 73.46 | 74.12 | **75.26** |
| | | 10% | 100% | 82.16 | 82.78 | 83.42 | 84.02 | **86.26** | 81.67 | 82.14 | 83.12 | 83.56 | **85.78** |
| | | 55% | 55% | 77.36 | 77.78 | 78.26 | 78.66 | **80.39** | 76.46 | 76.58 | 77.85 | 78.12 | **79.87** |
| Hateful Memes (AUROC) | 50% | 100% | 50% | 58.32 | 58.56 | 58.78 | 59.31 | **60.24** | 60.56 | 60.31 | 60.87 | 61.87 | **62.32** |
| | | 50% | 100% | 60.34 | 61.12 | 61.34 | 61.78 | **62.34** | 62.41 | 62.35 | 63.13 | 63.88 | **64.46** |
| | | 75% | 75% | 62.34 | 62.87 | 63.14 | 63.24 | **63.78** | 64.87 | 65.84 | 65.46 | 65.86 | **66.02** |
| | 70% | 100% | 30% | 58.54 | 59.02 | 59.36 | 60.02 | **60.56** | 60.74 | 61.12 | 61.26 | 61.56 | **62.82** |
| | | 30% | 100% | 60.12 | 60.78 | 61.32 | 61.54 | **62.32** | 62.74 | 63.24 | 63.14 | 63.48 | **64.12** |
| | | 65% | 65% | 62.34 | 62.56 | 63.12 | 63.32 | **63.78** | 64.82 | 65.04 | 65.23 | 65.48 | **66.08** |
| | 90% | 100% | 10% | 58.02 | 57.34 | 58.32 | 59.02 | **60.34** | 60.03 | 57.21 | 60.74 | 61.14 | **62.08** |
| | | 10% | 100% | 59.02 | 59.32 | 60.21 | 60.56 | **61.34** | 61.46 | 61.52 | 61.87 | 62.42 | **63.87** |
| | | 55% | 55% | 62.32 | 62.56 | 63.24 | 63.78 | **64.34** | 64.32 | 63.34 | 64.85 | 65.37 | **66.78** |

**Performance v.s. MMP when owning comparable parameters.** We compare our method with MMP by allowing them to own comparable parameters in Tab. 5. Specially, either when we decrease the required parameters of our method to 0.2M (0.2% of the entire model) by reducing the channel dimension, or we simultaneously increase the required parameters of both methods, our method consistently outperforms MMP, which verifies its effectiveness.

Table 5: Performance v.s. MMP when owning comparable parameters on the val set of MM-IMDb [2].

| Parameters | 0.2M | 1.6M | 2.8M | 4.0M | 5.2M | 6.4M |
|---|---|---|---|---|---|---|
| MMP(%) | 51.34 | 50.74 | 50.82 | 50.72 | 50.64 | 50.46 |
| Ours(%) | 53.14 | 53.56 | 53.88 | **54.24** | 54.12 | 53.98 |

**Limitations and broader impacts.** The limitations include (1) we only test the effectiveness upon two commonly-used two-stream multimodal models, and don't apply our method to other popular multimodal models. (2) we only include two modalities in the experiments, and will incorporate more modalities in the future. The broader impacts include (1) The missing-modality cases happen at times in real life. This paper proposes a novel method by adapting large multimodal models towards missing-modality scenarios, which increases the robustness of large multimodal models. (2) Our methods can notably decrease the required computations compared to previous methods upon missing-modality learning, which achieves a better accuracy-computation trade-off in real life.

## 5  Conclusion

In this paper, we tackle two main challenges in multimodal learning, including (1) any modality may be missing in any learning phase (e.g., training, testing or both), and (2) how to efficiently adapt large-scale multimodal transformers to fit missing-modality cases. We propose deep correlated prompting by leveraging the correlations between prompts of different layers and the relationships between prompts and input features. Results on three diverse datasets across various missing types and missing ratios verify the effectiveness of our proposed method.

## Acknowledgments and Disclosure of Funding

This work is supported by National Key Research and Development Program of China (2020YFC1522700) and National Natural Science Foundation of China (Project No. 62072334 and No. 62276182).

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

# A Appendix / supplemental material

**Compatibility with complete data.** In this paper, we have previously assumed that there is no guarantee modality-complete data can be entirely collected due to various factors in real-world scenarios. However, these still exist publicly-available modality-complete datasets which are carefully collected and annotated for model training. Thus, we perform additional experiments to compare our method with the baseline and input-level prompting by training on modality-complete data. To stimulate the missing-modality case during testing, we randomly select data with different modality-missing cases for each data pair (i.e., text-only, image-only, or complete) during training. Note that, different from other experimental settings introduced in the paper, here one data pair can be in various missing-modality cases at different training epochs. We plot the results in Fig. 5. It is observed that our method consistently outperforms the baseline and input-level prompting with better performance across different missing ratios.

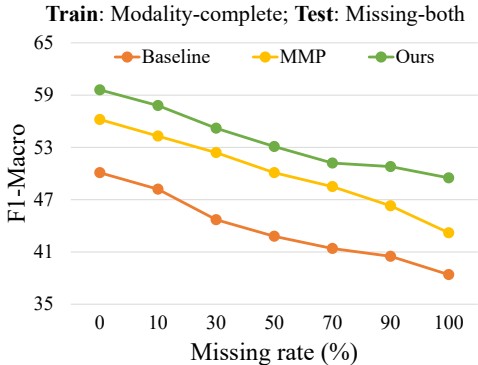

Figure 5: All models are trained with modality-complete data, where each data pair can be randomly assigned with different missing modalities (i.e., text-only, image-only, and modality-complete) at different training epochs to mimic the possible missing modalities during testing. Evaluation is on missing-both cases with different missing rates.

**Deployment on other backbones.** We deploy our proposed method upon other commonly-used vision-language models like CoCa [39] and ViLT [18], to verify its flexibility. CoCa is trained jointly with contrastive loss and captioning loss, which subsumes model capabilities from both contrastive approaches like CLIP [29] and generative methods like SimVLM [34]. ViLT is a single-stream multimodal backbone which directly concatenates the input text features and image features and feed them into a common transformer for processing. We adopt ViT-B/32 [7] as the image encoder for CoCa, and list the results in Table. 6 and Table 7, respectively. It is observed that our method can bring notable performance boost across three datasets compared to the baseline, which verifies its flexibility across various large-scale vision-language models.

Table 6: Quantitative results with CoCa [39] as our backbone on the MM-IMDB, UPMC Food-101, and Hateful Memes datasets with missing rate $\eta = 70\%$ under various modality-missing scenarios. The bold number indicates the best performance.

| Datasets | Image | Text | Baseline | Ours |
|---|---|---|---|---|
| MM-IMDb (F1-Macro) | 100% | 30% | 38.96 | **49.34** |
| | 30% | 100% | 45.78 | **55.12** |
| | 65% | 65% | 42.35 | **51.24** |
| Food101 (Accuracy) | 100% | 30% | 73.41 | **77.56** |
| | 30%% | 100% | 82.34 | **86.26** |
| | 65% | 65% | 77.24 | **80.46** |
| Hateful Memes (AUROC) | 100% | 30% | 54.87 | **58.36** |
| | 30% | 100% | 56.46 | **60.34** |
| | 65% | 65% | 57.82 | **61.83** |

Table 7: Quantitative results with ViLT [18] as our backbone on the MM-IMDB, UPMC Food-101, and Hateful Memes datasets with missing rate $\eta = 70\%$ under various modality-missing scenarios. The bold number indicates the best performance.

| Datasets | Image | Text | MMP | Ours |
|---|---|---|---|---|
| MM-IMDb (F1-Macro) | 100% | 30% | 39.22 | **45.26** |
| | 30% | 100% | 46.30 | **51.24** |
| | 65% | 65% | 42.66 | **48.45** |
| Food101 (Accuracy) | 100% | 30% | 74.53 | **78.85** |
| | 30% | 100% | 86.18 | **86.76** |
| | 65% | 65% | 79.08 | **80.85** |
| Hateful Memes (AUROC) | 100% | 30% | 59.11 | **61.24** |
| | 30% | 100% | 63.06 | **64.12** |
| | 65% | 65% | 66.07 | **66.68** |

Table 8: Ablations for the prompt length on the MM-IMDB dataset with missing rate $\eta = 70\%$. The bold number indicates the best performance.

| Prompt length | 12 | 24 | 36 | 48 | 60 |
|---|---|---|---|---|---|
| F1-Macro(%) | 53.56 | 53.88 | **54.24** | 54.12 | 53.98 |

**Ablations for the prompt length.** We ablate the prompt length for our DCP on the MM-IMDB dataset with missing rate $\eta = 70\%$ in Tab. 8. It's observed that the performance continues to increase when the prompt depth ranges from 12 to 36, and reaches a peak when the prompt depth equals 36. Further increasing thr prompt depth would degrade the performance. We thus set the prompt depth as 36.

