# OpenReview forum: "Deep Correlated Prompting for Visual Recognition with Missing Modalities"
_NeurIPS.cc/2024/Conference — NeurIPS 2024 poster_

### Official Review · Reviewer_YNgk · 2024-07-11

**Soundness:** 3
**Presentation:** 3
**Contribution:** 2
**Rating:** 5
**Confidence:** 4

**Summary:**

The paper proposes a prompt optimization approach to the missing modality issues in multimodal learning. Inspired by the missing-aware prompt (MMP), this paper adds more prompts, including correlated, dynamic and modal-common prompts, to each encoder to improve the performance. The experiment on three datasets shows the effectiveness of the proposed method.

**Strengths:**

The missing modality issue in multimodal learning is a practical challenge.

The designed method is clearly presented.

**Weaknesses:**

1. The novelty of the proposed method is limited since the MMP has proposed the prompt optimization approach to solving the missing modality issue. Compared with MMP, this paper adds more parameters in the form of prompt tokens from different inputs and functions.

2. The empirical comparison with MMP is probably not quite fair as the proposed method uses more additional parameters compared with MMP. According to Line 337, this method adds 2.4% additional parameters, while MMP only adds 0.2%.

**Questions:**

What is the specific contribution of this paper compared with MMP, other than adding more parameters and functions?


Will MMP's performance be better or comparable when MMP uses the same number of parameters as the proposed method?

**Limitations:**

Yes.

---

> ### Author Rebuttal · Authors · 2024-08-07
>
> Reviewer# YNgk
> 1. Novelty compared to MMP
>
>   Many thanks for your question. MMP has first introduced prompt learning to handle the missing-modality setting. It inserts learnable tensors, i.e., prompts at each layer which still keeping the image encoder and text encoder fixed to guide the model to fit missing-modality cases. However, the inserted prompts of MMP across different layers are independent, while we believe that the prompts across different layers and various modalities can provide beneficial information for each other to better fit missing-modality cases. Thus, we propose correlated prompts which generate the prompts of the next layer based on the prompts of both modalities in the current layer. For dynamic prompts, our intuition is that the prompts proposed by MMP are fixed for different inputs during inference and fail to fit the missing cases of different inputs, and we thus propose to dynamically compute the prompts based on different input features to better guide the behavior of the model. This procedure is implemented by a self-attention layer with a randomly initialized tensor as a query and the input features as keys and values. Besides, we propose modal-common prompts which store the shared information across different modalities, which can complement the model with common information across different modalities and facilitate the model to encode modal-specific information to better handle the missing scenarios in each modality.
>
> 2. Performance v.s. MMP when owning comparable parameters
>
> We compare our method with MMP by allowing them to own comparable parameters. Specially, either when we decrease the required parameters of our method to 0.2M (0.2% of the entire model) by reducing the channel dimension, or we simultaneously increase the required parameters of both methods, our method consistently outperforms MMP, which verifies its effectiveness.
>
> | Parameters| 0.2M | 1.6M | 2.8M | 4.0M | 5.2M | 6.4M |
>  | :---:|:---:|:---:|:---:|:---:|:---:|:---:|
>  | MMP | 51.34%| 50.74% | 50.82% | 50.72% | 50.64% | 50.46%|
>  | **Ours** | **53.14%** | **53.56%** | **53.88%** | **54.24%** | **54.12%** | **53.98%** |

---

> > ### Comment · Reviewer_YNgk · 2024-08-13
> >
> > Thanks for the response. My concern about the experimental comparison is addressed, so I will increase the score to 5. The novelty concern still holds but not a major flaw as the novelty concern is possibly not quite objective as it always is, so I wouldn’t fight for rejection.

---

### Official Review · Reviewer_BZuF · 2024-07-15

**Soundness:** 3
**Presentation:** 3
**Contribution:** 3
**Rating:** 6
**Confidence:** 4

**Summary:**

The model proposes prompting strategy where both modalities (image and text) are prompted, and the prompt for both modalities are correlated. The strategy is to use multiple prompts, namely correlated prompts, dynamic prompts, and modal-common prompts. As the backbone itself is multimodal (CLIP), it is a good idea to consider synchronized multi-modal prompts to fully harness the model capabilities when prompting it. The model surpasses multiple multimodal SoTAs on multiple datasets and also has proven to be effective in handling missing modalities in training and inference.

**Strengths:**

1. The strategy of using multiple types of multimodal prompts, along with the correlation strategy, is logically sound as the multimodal backbone itself is trained to understand the relationship between image and text modalities.

2. The modal surpasses multiple SoTAs on multiple benchmarks with considerable score improvement.

3. The ablation studies are sufficient to understand the justification of the network design.

**Weaknesses:**

1. Ablation studies regarding the multimodal backbone, e.g. using other model than CLIP or use dedicated unimodal encoders for each modality, highly recommended to increase paper quality.
2. In table 4, what are the performances when either image or text modalities are completely missing?

**Questions:**

1. Please elaborate further on how modal-common features are disentangled.
2. If possible, show the layer J-th in Figure 1 (framework overview)
3. Minor suggestion: The phrase "abundant ablations" in the introduction is a bit overboard, I suggest to write it as just "Ablation studies are further given..."

**Limitations:**

The limitations are discussed in the appendix, including that only text and visual modalities are tested with this model and the number of tested models.

---

> ### Author Rebuttal · Authors · 2024-08-07
>
> 1. Ablation studies by using other multimodal backbones
>
> We provide the results by comparing our method with the baseline method upon the single-stream ViLT backbone, and also comparing them upon the two-stream CoCa backbone as below.
>
> We first provide the results on the ViLT backbone. Our method large outperforms the baseline across different settings on three datasets.
>
> |  Datasets               | Image | Text  | MMP     | Ours    |
> |:---|:---:|:---:|:---:|:---:|
> |   MM-IMDb (F1-Macro)   | 100%  | 30%   | 39.22   |  **45.26**  |
>  | MM-IMDb (F1-Macro) | 30%                     | 100%  | 46.30 |  **51.24**  |
> | MM-IMDb (F1-Macro)  | 65%                     | 65%   | 42.66 |  **48.45**  |
> |  Food101 (Accuracy)  | 100%  | 30%   | 74.53   |  **78.85** |
> | Food101 (Accuracy)    | 30%                     | 100%  | 86.18 | **86.76**   |
> |Food101 (Accuracy)  | 65%                     | 65%   | 79.08 | **80.85**   |
> | Hateful Memes (AUROC)  | 100%  | 30%   | 59.11   |  **61.24**  |
> | Hateful Memes (AUROC)  | 30%                     | 100%  | 63.06 | **64.12**   |
> | Hateful Memes (AUROC)| 65%                     | 65%   | 66.07 | **66.68**   |
>
> We offer the results on the CoCa backbone. We can find that our method achieves superior performance than the baseline.
>
> |  Datasets               | Image | Text  | Baseline | Ours    |
> |:---|:---:|:---:|:---:|:---:|
> |  MM-IMDb (F1-Macro)     | 100%  | 30%   | 38.96    |  **49.34**  |
> |  MM-IMDb (F1-Macro) | 30%                     | 100%  | 45.78 |  **55.12**   |
> |  MM-IMDb (F1-Macro) | 65%                     | 65%   | 42.35 |  **51.24**   |
> |  Food101 (Accuracy)     | 100%  | 30%   | 73.41    |  **77.56**  |
> |  Food101 (Accuracy) | 30%                     | 100%  | 82.34 | **86.26**    |
> |  Food101 (Accuracy) | 65%                     | 65%   | 77.24 | **80.46**    |
> |  Hateful Memes (AUROC)  | 100%  | 30%   | 54.87    |  **58.36**  |
> |  Hateful Memes (AUROC) | 30%                     | 100%  | 56.46 | **60.34**    |
> |  Hateful Memes (AUROC) | 65%                     | 65%   | 57.82 | **61.83**    |
>
> 2. Performance when either image or text modalities are completely missing in Table 4
>
> We provide the results as follows.
> |  Datasets | Image | Text  |  CoOp  | MMP   | MaPLe | DePT  | Ours  |
> |:---|:---:|:---:|:---:|:---:|:---:|:---:|:---:|
> |  MM-IMDb (F1-Macro)     | 100%  | 0%    | 45.24  | 46.32 | 46.85 | 48.02 | **50.64** |
> |  MM-IMDb (F1-Macro) | 0%  | 100%  | 51.23 | 52.21  | 52.76 | 54.13 | **55.45** |
> |  Food101 (Accuracy)     | 100%  | 0%    | 70.34  | 71.52 | 71.18 | 72.25 | **73.87** |
> |  Food101 (Accuracy) | 0% | 100%  | 80.76 | 81.52  | 82.13 | 83.24 | **85.64** |
> |  Hateful Memes (AUROC)  | 100%  | 0%    | 59.56  | 60.14 | 60.25 | 61.32 | **62.12** |
> |  Hateful Memes (AUROC) | 0%   | 100%  | 60.87 | 61.32  | 61.54 | 62.12 | **63.24** |
>
> 3. How modal-common features are disentangled
>
>  Specifically, we introduce a shared learnable prompt across different modality encoders which embeds modal-common information for different modalities. To transform it into the feature space of various modalities, we introduce an independent projection function for each modality to change the channels of the shared learnable prompt. It explicitly embeds modal-common information and inversely encourages other prompts to provide modal-specific information to guide the model to handle different missing-modality cases.
>
> 4. Show the layer J-th in Figure 1 (framework overview)
>
> Thanks for your comments. We will update the J-th layer in our manuscript.
>
> 5. Phrase correction.
>
> Many thanks for your advice, we will update the expressions.

---

> > ### Comment · Reviewer_BZuF · 2024-08-12
> >
> > Thank you to the authors for addressing the questions. The authors have addressed most of my concerns, and I hope the answer can be further added in the final version of the paper either in the main paper or in the appendix. Overall, I stand with my initial rating of Weak Accept as the authors have demonstrated that their proposed method is competitive with logical explanation and proof to back it up.

---

### Official Review · Reviewer_64zT · 2024-07-16

**Soundness:** 4
**Presentation:** 3
**Contribution:** 3
**Rating:** 5
**Confidence:** 5

**Summary:**

This paper addresses the challenge of generalized missing modalities in multimodal learning, where a modality can be absent during any learning phase (e.g., training, testing, or both). he authors investigate prompt learning with missing modalities and propose deep correlated prompts designed to capture various types of correlations between prompts and input features across different modalities. Specifically, the proposed prompts include mechanisms for perceiving beneficial information from preceding layers, dynamically generating prompts based on input characteristics, and leveraging the complementary information from multimodal inputs. These designs improve the robustness of large multimodal models (e.g., CLIP) to missing modalities. Extensive experiments and ablation studies demonstrate consistently superior performance and verify the effectiveness of the proposed method.

**Strengths:**

1.	This paper addresses a more challenging missing modality setting, where modalities may be absent during both training and testing phases, making it highly practical and essential for real-world applications.
2.	The paper is well-motivated. The authors highlight the weaknesses of prior work and propose several designs (e.g., deep correlated prompts, dynamic prompts, common prompts) to improve robustness.
3.	The paper explores various types of correlations between prompts and input features across different modalities, and the proposed designs for each are technically sound.
4.	Extensive experiments show great improvement on the baseline and consistently superior performance compared to other methods across all benchmarks.
5.	Comprehensive ablation studies are conducted to validate the effectiveness of each proposed component.

**Weaknesses:**

1.	The paper lacks a detailed explanation or discussion on the efficacy of different prompt designs. In Figure 2, it shows that sequentially adding different designs improves the baselines, but it does not discuss the individual improvement gains for each design. Additional discussion on each design could help validate whether the increasing gains from sequentially adding designs are not merely due to more learnable parameters.
2.	The paper lacks visualization of each learnable prompt (e.g., deep correlated prompts, dynamic prompts, and common prompts). Visualizations could help validate whether the different components work as expected. For example, do dynamic prompts genuinely capture the different characteristics of inputs, or do they merely distinguish between different missing cases, which might be easier to learn due to the obvious absence of a modality?
3.	For each available modality, it seems there are a total of $(3*(2^M-1))$ prompts for each missing modality case. This could lead to an exponential increase and redundant prompts as more modalities are considered (i.e., M>2). For example, in a vision-and-language task, in the case of complete and missing-image, the text modality is available for both cases. However, it requires two separate prompt sets for the text encoder, which may actually learn the prompts for the same “text-available” case.

**Questions:**

1. In Table 4, I noticed that some values of the related work MMP are the same as the figures recorded in the paper. For example, the settings with:
   - missing rate = 70% (100% image and 30% text) in MM-IMDb,
   - missing rate = 70% (30% image and 100% text) in Hateful Memes,
   - missing rate = 70% (65% image and 65% text) in Hateful Memes.

   As far as I know, the MMP backbone model is the multimodal transformer ViLT. The authors state they re-implemented MMP on their setting (i.e., CLIP) for a fair comparison. It seems that the numbers should not be the same since they use different backbone models. Can the authors clarify why the values are identical despite using different backbone models?

2.	According to the design of prompts, it seems that the proposed method is not limited to two-stream models (e.g., it could be applied to single-stream models without using Eq. (5)). Generalizing the method to single-stream models and comparing it with related works could be helpful in verifying the generalizability of the proposed method. Have the authors tried it for single-stream models? If so, what were the results?
3.	I am willing to revise my rating if the authors also address the concerns mentioned in the weaknesses.

**Limitations:**

One limitation is that the proposed method requires modality-specific deep correlated prompts for each available modality, which could be challenging to extend to more modalities (e.g., five or more modalities).

---

> ### Author Rebuttal · Authors · 2024-08-07
>
> 1. Efficacy of each proposed prompt.
>
>   We place each proposed prompt upon the baseline method, and show the results as below on the MMIMDb dataset upon the missing-both setting with η=70%. It’s observed that each proposed prompt could notably boost the performance.
>   | Configurations| Extra brought parameters | F1-Macro|
> | --- | --- | --- |
> | Baseline | - | 49.21% |
> | +Correlated prompts | +1.2M| 52.12% |
> | +Dynamic prompts | +1.8M | 51.04% |
> | +Modal-common prompts | +1.0M | 51.26% |
>
> 2. Visualization of each learnable prompt.
>
>   We offer visualizations for the dynamic prompts using the T-SNE method on the Food101 dataset upon the missing-both setting with η=70% and η=50%. As Food101 have 101 classes which is redundant for visualization, we select a subset with 11 classes to give better visualization results. It’s observed that the dynamic prompts could roughly categorize the prompts into distinct classes. Besides, for different missing settings with η=70% and η=50%, the distribution of dynamic prompts with respect to the same inputs are different, which show that dynamic prompts can learn to generate various expressions for each input upon different missing settings.
>
> 3. Exponential increased prompts when more modalities are introduced.
>
>  For M input modalities, it requires $2^M-1$ types of prompts for all missing-modality cases. This may be redundant, but we expect a specific prompt to handle each missing-modality case to better guide the model to handle different missing scenarios. Besides, the overall extra consumed parameters are quite few as the modalities increase as the projection functions are shared across different modalities. Thus as the modalities increase, we only introduce new learnable prompts which occupy quite few parameters. For example, for each missing-modality case, the newly introduced prompts own 768*36=27648≈0.03M parameters with a prompt length of 36 for each modality encoder.
>
> 4. Identical values in Table 4 compared with MMP
>
> Sorry for the errors. We wrongly set the values when organizing the tables from different resources. The correct values should be 48.23 for missing rate = 70% (100% image and 30% text) in MM-IMDb, 61.12 for missing rate = 70% (100% image and 30% text) in Hateful Memes, and 63.24 for missing rate = 70% (30% image and 100% text) in Hateful Memes. We will current them in the manuscript.
>
> 5. Results on single-stream backbones
>
>   We compare our method with MMP based on the ViLT backbone. We don’t use features of two modalities to generate prompts of the next layer as Eq.5. The results on three datasets with η=70% upon different missing settings are shown below. It’s observed that our method shows superior performance than MMP.
>
>
> |  Datasets               | Image | Text  | MMP     | Ours    |
> |-------------------------|-------|-------|---------|---------|
> |   MM-IMDb (F1-Macro)   | 100%  | 30%   | 39.22   |  **45.26**  |
>  | MM-IMDb (F1-Macro) | 30%                     | 100%  | 46.30 |  **51.24**  |
> | MM-IMDb (F1-Macro)  | 65%                     | 65%   | 42.66 |  **48.45**  |
> |  Food101 (Accuracy)  | 100%  | 30%   | 74.53   |  **78.85** |
> | Food101 (Accuracy)    | 30%                     | 100%  | 86.18 | **86.76**   |
> |Food101 (Accuracy)  | 65%                     | 65%   | 79.08 | **80.85**   |
> | Hateful Memes (AUROC)  | 100%  | 30%   | 59.11   |  **61.24**  |
> | Hateful Memes (AUROC)  | 30%                     | 100%  | 63.06 | **64.12**   |
> | Hateful Memes (AUROC)| 65%                     | 65%   | 66.07 | **66.68**   |

---

> > ### Author Response · Authors · 2024-08-13
> > **Response for Reviewer 64zT**
> >
> > Dear reviewer, thanks for your time and efforts in reviewing our manusript. We have provided a point-to-point response regarning your concerns and we are looking forward to receiving your valuable feedback on the points we addressed in the response. If you have further concerns, place let us know and we will respond to you as soon as possible. Thank you for your dedication to the review process.
> >
> > Sincerely,
> > Authors

---

### Official Review · Reviewer_xVBv · 2024-07-17

**Soundness:** 3
**Presentation:** 3
**Contribution:** 2
**Rating:** 4
**Confidence:** 5

**Summary:**

This paper proposes to address the missing modality problem for the multimodal recognition model (i.e. the multi-modal data could be incomplete). There are three techniques of prompting being proposed (while the recognition model, i.e. two-stream multimodal method CLIP in this paper, is kept fixed), including: 1) correlated prompts, where a part of the prompts in the input-level are firstly selected according to the missing scenario (e.g. complete, text-only, or image-only), then the prompt in each of the following network layers are predicted from the multimodal prompt of its preceding layer; 2) dynamic prompts, the input-level prompts contain a portion generated according to the input sample; 3) modal-common prompts, where the rest of the input-level prompts is stemmed from a common component shared across modalities. The combination of the aforementioned three techniques experimentally shows better performance in comparison to various baselines (mainly the SOTA method from MMP [17]).

**Strengths:**

+ The proposed method provides superior performance with respect to various baselines and its proposed techniques (i.e. correlated prompts, dynamic prompts, modal-common prompts) are experimentally shown to benefit the model performance.
+ The extensive experiments are conducted on multiple dataset with various experimental settings.
+ The presentation is clear and easy to follow.

**Weaknesses:**

- The modal-common prompts and the dynamic prompts actually are not directly connected to the missing modality problem (or being irrelevant to different cases of missing modality). While excluding these two prompting techniques from the proposed method (in which such variant becomes "Ours (A)" in Figure 2), the improvement with respect to the state-of-the-art approach of handling missing modality (i.e. MMP[17]) would become marginal (please include MMP[17] into the ablation study shown in Figure 2 or directly provide the tabular quantitative results for the ablation study). Similarly, while we only consider the technique of correlated prompts as the manner in the proposed to tackle the missing modality, it becomes the only difference in the proposed method compared to MMP [17] (in terms of methodology), thus leading to the concern of limited novelty. Furthermore, there should be a baseline of integrating the modal-common prompts (acting as a basic component of prompt) and dynamic prompts into MMP[17] to better highlight the contribution of the proposed correlated prompting technique (which is the main technique in the proposed method to be connected with the missing modality challenge). Moreover, as modal-common prompts and the dynamic prompts introduce additional learnable parameters (in comparison the correlated prompts), there should be further detailed analysis/comparison in terms of number of learnable parameters versus model performance.
- Though the proposed dynamic prompts do experimentally shown to improve the overall performance under various missing modality cases, such prompting technique is actually not new, where we can see its similar application in various research problems (e.g. Wu et al., IDPG, NAACL'22; Lu et al., PromptPG, ICLR'23; Qiu et al., FedTPG, FL@FM-NeurIPS’23).

**Questions:**

Although currently the proposed method seems to provide superior performance with respect to various baselines and its proposed techniques (i.e. correlated prompts, dynamic prompts, modal-common prompts) are experimentally shown to benefit the model performance, there are concerns regarding limited novelty (where only the correlated prompts are considered to be related to missing modality while the other two techniques, i.e. dynamic and modal-common prompts, are not) and detailed analysis for the number of learnable parameters versus model performance, (as listed in the weaknesses), in which the the authors are highly encouraged to make the corresponding clarifications in the rebuttal.

**Limitations:**

no potential negative societal impact is found.

---

> ### Author Rebuttal · Authors · 2024-08-07
>
> 1. Proposed prompts not directly connected to the missing modality problem.
>
> Sorry for the mis-clarification in the manuscript to mislead you. In line 146-150 of our manuscript, we state that we set different prompts for various missing modalities. Specifically, for correlated prompts, we independently set the initial prompt of the first layer for different missing-modality settings, which will be transformed via a projection function to generate the prompts for the next layer. For the dynamic prompts, we initialize different queries for different missing modalities, which are then used to compute the dynamic prompts based on input features in a self-attention manner. For the modal-common prompts, we initialize different modal-common prompts for different missing-modality settings, which will be transformed into embedding space via a projection function to provide prompts. Sorry for missing the expressions for the dynamic prompts and modal-common prompts upon different missing-modality settings. We will update them in the manuscript to give better clarifications.
>
> 2. Effectiveness regarding the correlated prompts compared to MMP.
>
> Based on the same CLIP backbone, we compare the performance of MMP to our proposed correlated prompts on the MMIMDb dataset with different missing ratios under the missing-both case as below. It’s observed that our proposed correlated prompts consistently offer superior performance than MMP under the same setting.
>
> | Settings| η=10% | η=30%| η=50% | η=70% | η=90% |
> | --- | --- | --- | --- | ---| ---|
> | MMP | 56.54%| 53.64% |52.12%| 51.34% | 48.04%|
> | Correlated prompts | **57.91%** | **54.84%** | **53.78%** | **52.12%** | **48.52%** |
>
> 3. Comparison between the integration of MMP and ours.
>
>   We integrate the proposed dynamic prompts and modal-common prompts into MMP based on the CLIP backbone to form a method termed as MMP*, and compare our method with it on the MMIMDb dataset with different missing ratios under the missing-both case as below. It’s observed that our method notably outperforms MMP* with different η.
>
> | Settings| η=10% | η=30%| η=50% | η=70% | η=90% |
> | --- | --- | --- | --- | ---| ---|
> | MMP* | 58.58%| 55.46% |54.16%| 52.96% | 50.04%|
> | Ours | **59.82%** | **56.87%** | **55.45%** | **54.24%** | **51.26%** |
>
> 4. Analysis between parameters v.s. performance
>
>   We first give an ablation for the parameters of each proposed prompt. We give the performance with η=70% on the MMIMDb dataset upon the missing-both setting. As shown below, as we insert each proposed prompt, the performance consistently increases. The proposed prompts overall bring extra 4.0M trainable parameters, which is only 2.4% of the overall framework.
>
>  | Configurations| Brought parameters | F1-Macro|
> | --- | --- | --- |
> | Baseline | - | 49.21% |
> | +Correlated prompts | +1.2M| 52.12% |
> | +Dynamic prompts | +1.8M | 53.22% |
> | +Modal-common prompts | +1.0M | 54.24% |
>
>   Besides, we test the relationships between the brought extra parameters and performance, and show the results as below. It’s observed that the performance continues to increase when the prompt depth ranges from 12 to 36, and reaches a peak when the prompt depth equals 36. The parameters consistently increase as the prompt length grows.
>
> | Brought parameters | 1.6M | 2.8M | 4.0M | 5.2M | 6.4M|
> | --- | --- | --- | --- | --- | --- |
>  | F1-Macro | 53.56% | 53.88% | **54.24%** | 54.12% | 53.98% |
>
>  Finally, we compare our method with MMP with the same parameters. It’s observed that with the same parameters, our method consistently achieves better performance.
>
> | Parameters| 0.2M | 1.6M | 2.8M | 4.0M | 5.2M | 6.4M |
>  | --- | :---: | :---: | :---: | :---: | :---: | :---: |
>  | MMP | 51.34%| 50.74% | 50.82% | 50.72% | 50.64% | 50.46%|
>  | Ours | **53.14%** | **53.56%** | **53.88%** | **54.24%** | **54.12%** | **53.98%** |
>
> 5. The dynamic prompting technique is not new.
>
>   Many thanks for your question. The dynamic prompts have been previous investigated in other methods. However, dynamic prompts in the missing-modality scenarios have not been studied. It’s worth exploring whether generating various prompts according to different missing cases and input features can well guide the model to fit the missing-modality settings. Our manuscript verifies the efficacy of this design.

---

> > ### Author Response · Authors · 2024-08-13
> > **Response for Reviewer xVBv**
> >
> > Dear reviewer, thanks for your time and efforts in reviewing our manusript. We have provided a point-to-point response regarning your concerns and we are looking forward to receiving your valuable feedback on the points we addressed in the response. If you have further concerns, place let us know and we will respond to you as soon as possible.Thank you for your dedication to the review process.
> >
> > Sincerely,
> > Authors

---

### Official Review · Reviewer_nnhg · 2024-07-17

**Soundness:** 2
**Presentation:** 2
**Contribution:** 2
**Rating:** 6
**Confidence:** 4

**Summary:**

This paper proposes a new method to handle missing modalities in visual and language recognition systems.
The paper proposes a very similar method to the one proposed by MMP [17] but using different way of getting the prompts to feed them into the transformer layers.
Comparison with other works show that the method seems to be effective and some ablations studies are performed to study the different design choices. The method is validated using the most common datasets for this task.

**Strengths:**

- The method seems to work when compared with other state-of-the-art models.
- The paper presents results on several datasets and with different settings of the model.

**Weaknesses:**

- The main weakness of the paper is clarity. There are three different sets of prompts that are appended to the intermediate representations. However, the only difference between them seems to be the type of architecture the method uses to compute them. The explanation is very limited and Figure 1 does not illustrate where do these prompts come from. Without the clarity of this explanation it becomes really hard to understand how the motivation of each type of prompt fits the design. What are exactly correlated prompts, dynamic prompts, and modal-common prompts? What make them correlated, dynamic and modal-common? This is not clear in the paper at all.

- It is not clear what is baseline. What does dropping features when modality is missing? The input sequence become shorter and coming from only a single modality? If that's the case, what is trainable and what is not?
Please explain well this part. I would expect that this baseline is: training with the same number of parameters as the base method, by simply adding learnable prompts at each layer and training using mod-drop (dropping modalities randomly when training, dropping modalities can be done by inputting noise instead of tokens, the average of the modality tokens, zeroes, or not passing the missing modality at the input, it is a design choice that needs to be explained). If it is not what I'm thinking, please explain well, since this is a key experiment.

- When comparing with MMP, how did the authors do it? Please explain exactly how was this re-implementation. Also, to be fair, the authors should have applied their method using ViLT instead of CLIP, in that way there is no doubt that this method is better than the very similar MMP.

- What is the zero-shot performance of CLIP on these datasets?

**Questions:**

- Please explain well the mechanism of the different types of prompts, input, output at train and test time for each one of them. It could have been done easily with a figure, but at least with a few sentences it could become clearer.
- What makes a "dynamic" prompt "dynamic"?
- What does baseline mean and how was implemented?
- How was MMP implemented on your framework?
- What if using ViLT instead of CLIP, would still your method be better than MMP?
- What is the zero-shot performance of CLIP on these datasets? it is important since this might be a robust method that does not suffer from missing modality. It can be implemented using nearest neighbor to each of the class embedding using either modality, and combining them when both are present.

**Limitations:**

Limitations have been addressed.

---

> ### Author Rebuttal · Authors · 2024-08-07
>
> 1. The mechanism of different prompts.
>
>   Many thanks for your question. We have plotted a figure to further illustrate our proposed prompts by comparing them with our baseline and MMP[17], which can be found in the pdf file of Author Rebuttal. The baseline simply uses fixed image encoder and text encoder and only finetunes the classifier to handle downstream tasks. To well adapt to missing modalities, MMP[17] inserts learnable tensors, i.e., prompts at each layer which still keeping the image encoder and text encoder fixed to guide the model to fit missing-modality cases. However, the inserted prompts of MMP across different layers are independent, while we believe that the prompts across different layers and various modalities can provide beneficial information for each other to better fit missing-modality cases. Thus, we propose correlated prompts which generate the prompts of the next layer based on the prompts of both modalities in the current layer. For dynamic prompts, our intuition is that the prompts proposed by MMP are fixed for different inputs during inference and fail to fit the missing cases of different inputs, and we thus propose to dynamically compute the prompts based on different input features to better guide the behavior of the model. This procedure is implemented by a self-attention layer with a randomly initialized tensor as a query and the input features as keys and values. Besides, we propose modal-common prompts which store the shared information across different modalities, which can complement the model with common information across different modalities and facilitate the model to encode modal-specific information to better handle the missing scenarios in each modality.
>
> 2. What makes a "dynamic" prompt "dynamic"?
>
> We propose to dynamically compute the prompts based on different input features, which avoids employing fixed prompts for different input features during training and inference. This procedure is implemented by a self-attention layer with a randomly initialized tensor as a query and the input features as keys and values. You can view the illustration in the figure of Author Rebuttal.pdf.
>
> 3. The baseline setting.
>
> The baseline setting is using the text encoder and image encoder to encode the input texts and images, whose output features are fed into the classifier for recognition. In this procedure, only the classifier is updated and other model components including the text encoder and the image encoder are kept fixed. The only difference between our method and the baseline is inserting learnable prompts at each layer which only bring few extra parameters. Specifically, when a modality is missing, we simply don’t feed the corresponding features into the modality encoder for processing and set the outputs of this modality encoder as zeros. Overall, we have tried three baseline settings, including (1) inputting zeros for the modality encoder when a modality is missing, (2) inputting the average of the modality tokens for the modality encoder when a modality is missing, and (3) not feeding input features for the modality encoder when a modality is missing and setting the outputs of this modality encoder as zeros (Default). As shown below, out experiments on the MMIMDb dataset with η=70% show the last choice gives best performance, and thus we adopt it as a stronger baseline.
>
> | Baseline Configurations | Acc(%) |
> | --- | :---: |
> | Inputting zeros | 47.65 |
> | Inputting averaged tokens | 46.24|
> | Default| **49.21**|
>
>
> 4. The re-implementation setting of MMP.
>
> MMP adds learnable prompts with a length of 16 at the bottom 6 layers based on the ViLT backbone. we re-implement it based on the CLIP backbone by adding learnable prompts with a length of 16 at the bottom 6 layers in the image encoder and text encoder, respectively. Our method owns the same number of inserted layers and prompt length with MMP. The only difference is that we insert three proposed prompts into the model.
>
> 5. Comparison with MMP using the ViLT backbone
>
> We compare our method with MMP using the ViLT backbone and show the results on three datasets with η=70% upon the missing-both setting as below. It’s observed that our method shows superior performance than MMP. It’s also noticed that using ViLT as the backbone achieves inferior performance than CLIP, and thus we adopt CLIP as the default backbone.
>
> | Dataset | MMP| Ours |
> | --- | :---: | :---: |
> | MMIMDb | 42.66% | **48.45%** |
> | Food101 | 79.08%| **80.85%** |
> | Hateful Memes | 66.07% | **66.68%** |
>
> 6. Zero-shot performance of CLIP.
>
>   We test the performance of zero-shot CLIP on the MMIMDb, Food101 and Hateful Memes datasets. We first calculate and store the averaged embeddings of all classes of both modalities on each dataset. When the input modalities are complete, we calculate the similarities between the output features and the pre-computed class embeddings for each modality encoder. We select the class with the highest similarity as the recognition output within both modalities. When a modality is missing, we select the class whose embedding owns the highest similarity with the output features of the available input modality as the recognition output. We compare the performance of zero-shot CLIP, finetuned CLIP (our baseline, which sets output features as zeros when a modality is missing) and our method as below. The zero-shot CLIP achieves inferior performance than the other methods, and ours perform best. We suppose that the zero-shot CLIP not finetuned on the downstream datasets fails to well adapt to downstream setting. Finetuning the CLIP could notably increase the performance, and ours further boost the performance by injecting different kinds of learnable prompts.
>
> | Datasets | Zero-shot CLIP | Finetuned CLIP | Ours |
> | --- | :---: | :---: | :---: |
> | MMIMDb | 34.52% | 49.21% | **54.24%**|
> | Food101 | 57.02% | 77.74% | **82.38%** |
> | Hateful Memes | 55.23% | 62.58% | **66.08%** |

---

> > ### Comment · Reviewer_nnhg · 2024-08-11
> >
> > Thanks to the authors for the thorough reply. I think the figure makes the contribution of the paper way more clear, and I suggest that it is included in the main manuscript.
> >
> > It is still not 100% clear to me what is baseline for you. I apologize for not being clear with my question in the first place.
> > **Fine-tuned CLIP** in the last Table that you showed in the rebuttal is the same baseline model that you used for Figure 2 and Figure 3 in the main paper? Was this baseline trained with missing modalities at train time (as a sort of "augmentation"), i.e. modality dropping at train time?
> >
> > That's my only remaining question, in order for me to give my final evaluation on the paper.

---

> > > ### Author Response · Authors · 2024-08-11
> > > **Response for Reviewer nnhg**
> > >
> > > Many thanks for your reply. We will try to save space to include the figures in our manuscript. For the baseline, the Fine-tuned CLIP in the last Table in the rebuttal is the same baseline model that we used for Figure 2 and Figure 3 in the main paper. The baseline is trained with the same missing modalities at training time to keep fair comparison with other methods. We also keep the identical training settings for other methods to make a fair comparison.

---

> > > > ### Comment · Reviewer_nnhg · 2024-08-11
> > > >
> > > > Authors have addressed most of my concerns. I still think the paper could be clearer at explaining the different model components, and be more thorough when explaining the different setups for baselines and other methods. The included experiments in the rebuttal helped clarifying most of the doubts. However, the paper will need a substantial effort at explaining better all the unclear points that I and other reviewers have raised. I will increase my score to 6, and I hope the authors work on the manuscript to improve its clarity and especially its reproducibility in terms of experiments, experimental setup, baseline definition and definition of model components. Thank you for your great work in this rebuttal.

---

> > > > > ### Author Response · Authors · 2024-08-11
> > > > > **Response for Reviewer nnhg**
> > > > >
> > > > > Many thanks for your careful comments. We will incorporate detailed explanations in the manuscript to clarify the improvements of our method compared to the baseline and existing methods. We will also make the code publicly available.

---

### Author Rebuttal · Authors · 2024-08-07

We provide (1) a figure to further illustrate out proposed three prompts by comparing them with our baseline and MMP[17]. (2) Visualizations for the dynamic prompts using the T-SNE method on the Food101 dataset upon the missing-both setting with η=70% and η=50%.

---

### Decision · Program_Chairs · 2024-09-25

**Decision:**

Accept (poster)

**Comment:**

The paper receives four positive and one negative ratings. Overall, the reviews initially have concerns about technical clarity (e.g., prompt designs, baseline setting) and experimental results (e.g., results using ViLT, contribution of each designed prompt, results of completely missing one modality). After the rebuttal, most reviewers find the major issues resolved (three reviewers upgraded the rating). The remaining concern from reviewer xVBv and YNgk is on the technical novelty. The AC took a close look at the paper and finds that designing these prompts is of great interest in the missing-modality situations, and hence recommends the acceptance decision. It is highly recommended that the authors should incorporate reviewers' suggestion in the paper (especially the illustration of prompt design, more experimental results) and release the code for fostering this important research area.